# Mitigation of Air Pollutants by UV-A Photocatalysis in Livestock and Poultry Farming: A Mini-Review

Myeongseong Lee [1], Jacek A. Koziel [2,3,*], Peiyang Li [3] and William S. Jenks [4]

1   Department of Animal Science, Texas A&M University, College Station, TX 77843, USA; leefame@tamu.edu
2   USDA-ARS Conservation and Production Research Laboratory, Bushland, TX 79012, USA
3   Department of Agricultural and Biosystems Engineering, Iowa State University, Ames, IA 50011, USA; peiyangl@iastate.edu
4   Department of Chemistry, Iowa State University, Ames, IA 50011, USA; wsjenks@iastate.edu
*   Correspondence: jacek.koziel@usda.gov; Tel.: +1-806-356-5744

**Abstract:** Ultraviolet (UV)-based photocatalysis has been the subject of numerous investigations focused on mitigating undesirable pollutants in the gas phase. Few works report on applications beyond the proof of the concept. Even less is known about the current state of the art of UV photocatalysis in the context of animal agriculture. A growing body of research published over the last 15 years has advanced the knowledge and feasibility of UV-A photocatalysis for swine and poultry farm applications. This review paper summarizes UV-A photocatalysis technology's effectiveness in mitigating targeted air pollutants in livestock and poultry farms. Specifically, air pollutants include odor, odorous VOCs, $NH_3$, $H_2S$ and greenhouse gases ($CO_2$, $CH_4$, $N_2O$). We trace the progression of UV-A photocatalysis applications in animal farming since the mid-2000 and developments from laboratory to farm-scale trials. In addition, this review paper discusses the practical limitations and outlines the research needs for increasing the technology readiness and practical UV application in animal farming.

**Keywords:** environmental catalysis; ultraviolet light; air purification; air pollution control; indoor air quality; odor; volatile organic compounds; greenhouse gases; ammonia; hydrogen sulfide

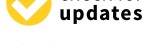

## 1. Introduction

As the scale of the livestock industry has grown with the increase in the demand for livestock and poultry products, gaseous emissions, an unwanted side effect of livestock and poultry production, are also increasing [1]. Gaseous emissions from livestock and poultry barns include components known collectively as greenhouse gases (GHGs, e.g., $CO_2$), volatile organic compounds (VOC, e.g., formaldehyde), along with others not typically included in those groupings, such as hydrogen sulfide ($H_2S$) and ammonia ($NH_3$). Other noxious components of the exhausts include the particulate matter (PM) of various sizes and airborne microbes. Cumulatively, these have a detrimental effect on human health, environment, climate and quality of life in rural communities [2–6].

Various mitigation technologies have been developed to reduce such air pollution, and the mitigation technologies are divided mainly into "source-based type", (meant to fundamentally reduce the emissions) and "end-of-pipe type" (physicochemical and biological treatment of the output from barns to reduce the release into the environment) [7]. Interestingly, ultraviolet light (UV) can be considered as both end-of-pipe (treating exhaust air from barns) and source-based type (treating air inside the barn) [8,9].

UV treatment technology was tested in the early 2000s because of its easy application and expected mitigation effects in barns. The UV spectrum is divided into four ranges because the energy of each photon is inversely related to its wavelength; different materials absorb and react in each range. Traditionally, the ranges are known as "A" (315–400 nm), "B" (280–315 nm), "C" (200–280 nm) and "vacuum" (100–200 nm) [10]. The term "vacuum"

arises because those wavelengths are absorbed by the components of ordinary air, and thus, the transmission of the light requires a vacuum. Although each of these causes photodamage to humans and animals on exposure, in broad terms, ultraviolet light-A (UV-A) is the least toxic, and vacuum ultraviolet (V-UV) is the most toxic. For example, UV-A is used in the commercial tanning industry, while standard germicidal bulbs are in the ultraviolet light-C (UV-C) range. In a variety of related pollutant mitigation applications, the most typical combination is the use of UV-A with a semiconductor photocatalyst that absorbs the light, causing the formation of reactive intermediates that in turn degrade the unwanted pollutants.

The photocatalysis reaction is initiated when photons of sufficient energy (more than the bandgap) are absorbed by the photocatalyst, resulting in electron ($e^-$)/hole ($h^+$) pair generation [11–14]. Nanosized titanium dioxide ($TiO_2$) is commonly applied to surfaces as a catalyst material, and it is effective with all UV with wavelengths below 400 nm [11,12,15–20]. Although the detailed mechanism of photocatalysis varies with different target pollutants, it is commonly agreed that the primary reactions responsible are interfacial redox reactions of electrons and holes with adsorbed pollutants or mediators, such as water [12,21,22].

The applicability of UV-A photocatalytic technology to the farm has been evaluated for mitigating odorous gases and fine particulate matter concentrations, as well as for increasing the feed conversion rates [1,8,23–32]. Relatively few works report on applications beyond the proof of the concept. Less is known about the current state of the art of UV-A photocatalysis in the context of animal agriculture. The purpose of this review paper is (1) to present the UV-A photocatalyst concept, (2) to summarize UV-A photocatalysis technology's effectiveness in mitigating targeted air pollutants in livestock and poultry barns, (3) to evaluate the potential of UV-A photocatalysis and (4) to suggest improvement recommendations for UV-A photocatalysis. We trace the progression of UV-A photocatalysis applications in animal farming since the mid-2000s and developments from laboratory to farm-scale trials. In addition, we discuss some practical limitations and outline research needs for increasing the technology readiness and practical UV-A photocatalytic application in animal farming.

## 2. Mechanism of UV-A Photocatalysis

Nanosized titanium dioxide ($TiO_2$) is commonly used as a photocatalyst material due to its relatively efficient photoactivity, high stability and lowest cost in the industry [11,12,15,17]. There are $TiO_2$ crystalline forms of anatase, rutile and brookite, but anatase and rutile of $TiO_2$ are widely used commercially due to high photocatalytic activity [12,16,17]. Activation of $TiO_2$ is initiated at wavelengths <400 nm [18–20].

Photocatalysis is, by definition, the acceleration of a photochemical reaction in the presence of a catalyst, typically, but not exclusively, with UV-A light. The primary photochemical event caused by the absorption of a photon (Figure 1) is the promotion of an electron from the valence band to the conduction band of the catalyst, causing charge separation, also frequently referred to as producing a valence band "hole" ($h^+$) and conduction band electron ($e^-$) [12–14,16,17,21,22,33]. At its most basic level of description, holes ($h^+$) react with water molecules to generate hydroxyl radicals ($\cdot OH$) or oxidize organic materials through direct reaction [13]. The electrons generated in the conduction band react with molecular oxygen ($O_2$) to form superoxide ions ($O_2^{\bullet-}$) [13,14,17]. Although a complete description of the cascade of reactions is complex, the photon's energy is used to convert an oxidatively neutral surface to one with strong oxidizing power and modest reducing power. In the absence of water, $O_2$ and other substrates, the conduction band electron will collapse back to the valence band, "wasting" the absorbed energy in the form of heat release. These processes are qualitatively illustrated in Figure 1.

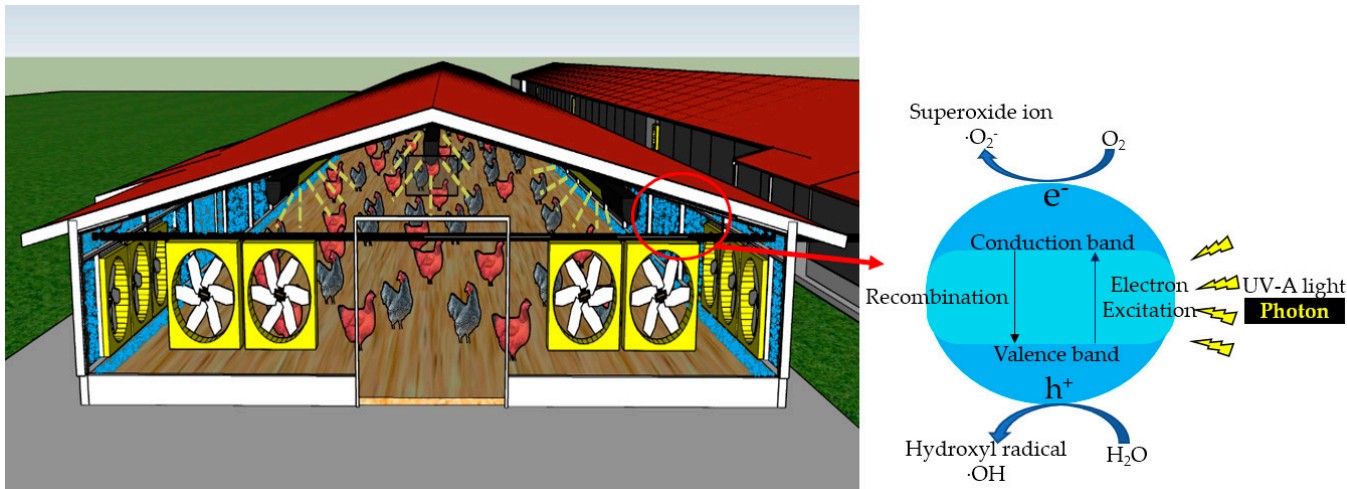

**Figure 1.** Mechanism of UV-A photocatalysis in livestock and poultry barns. A conceptual illustration of barn walls treated with photocatalyst (blue) irradiated with UV light (yellow rays). Note that UV-A light needs to be shielded, or the lighting duration needs to be controlled, as UV-A overexposure has side effects.

It is broadly believed that hydroxyl radicals (·OH) formed through photocatalysis can break down organic matter and gas emissions, with some contribution by the superoxide [14,17]. The decomposed gases from livestock and poultry barns will create by-products, as shown in Table 1. In most instances, the substrate is oxidized and is "mineralized" to the terminal oxide (e.g., $CO_2$ for carbon sources) on exhaustive treatment, but partial oxidation can result from less extensive treatment. Examples are given in Table 1.

**Table 1.** Summary of reported by-products after UV photocatalysis. This table was created by modifying the table presented in a previous report [29] published under the terms of CC BY.

| Reference | Target Gases | Reported Products after Photocatalysis |
|:---:|:---:|:---:|
| [34–37] | $NH_3$ | $N_2O$ and $N_2$ |
| [38–41] | $H_2S$ | $SO_2$ and $SO_4^{2-}$ |
| [42–45] | $CH_4$ | $CH_3OH$, H, $C_2H_6$, $C_2H_4$, $HCO_2H$ and $CO_2$ |
| [46,47] | $CO_2$ | $CH_4$, $CH_3OH$, HCHO and CO |
| [48,49] | $N_2O$ | $N_2$ and $O_2$ |
| [50] | $O_3$ | $O_2$ |
| [51] | VOCs | Partially oxidized species, $CO_2$ and $H_2O$ |

## 3. The Concept of UV Dose for UV-A Photocatalysis Evaluation

In determining the sufficient quantity of light in a particular apparatus, previous studies reported only light intensity (irradiance, $W \cdot m^{-2}$), but the actual light dosage is more complicated. At its most basic level, it is clear that a larger light dose is received when the residence time of a sample is raised, all other factors being held equal. For irradiation chambers of identical geometry and irradiance, we can reasonably approximate that the actual dosage is inversely related to the gas flow rate through the chamber. Additionally, complicating the matter is the fact that it is reasonable to assume that most of the reactions occur at or very near the photocatalyst surface; this implies that the physical geometry of the experimental setup will also have measurable effects on the rate/efficiency of degradation. Thus, one needs to be careful in directly comparing the quantitative results among reports that use very different irradiation chambers.

In order to compare the photocatalytic mitigation effect of previous studies conducted under different experimental settings and conditions, we use the concept of UV dose onto a catalyst-coated surface, as defined by Bolton and Linden [10], with units of J·m$^{-2}$ (or, for convenience, mJ·cm$^{-2}$). The Bunsen–Roscoe law, or the law of reciprocity [52], can be used to calculate the UV dose for a given apparatus Equation (1).

$$D = I \times t \tag{1}$$

where D is the UV dose (mJ·cm$^{-2}$), I is the irradiance or light intensity in mW·cm$^{-2}$, and t is the UV treatment time (or residence time in a flowing sample) in seconds.

The light intensity can be measured directly with a calibrated radiometer with a UV-A detector placed on the catalyst-coated panels. The dosage time is estimated by dividing the total volume of the chamber (m$^3$) by the volumetric airflow rate (m$^3$·s$^{-1}$) inside the chamber used in each experiment.

Although the UV dosage concept is a step forward for quantitative comparison, it alone may not be sufficient to allow direct comparison of quantitatively quoted efficiencies (or rates) from one laboratory to another. Given the results (see below) that most pollutants of interest are only partly removed, the limiting factor may also be the gas contact time with the catalyst surface, which could be a function of the geometric characteristics of an experimental design. Even the type or quantity of TiO$_2$ could affect quantitative comparisons. However, when the same chambers are used for different measurements, we believe the dosage is a good guide for quantitative comparison.

The difficulty of quantitative comparisons does not take away from the value of qualitative comparisons of experiments being carried out by different groups or using different chambers, especially when making broad conclusions about whether UV-A photocatalysis is at least a plausible strategy for pollutant mitigation. However, for clarity of comparison, all previous experimental conditions and UV doses we calculated from previous studies are reported in Table 2. In the subsequent tables describing the results, the reader is encouraged to use the references listed in each entry to see the description of experimental conditions given here.

**Table 2.** Summary of the experimental conditions and UV dose in previous research.

| Reference | Experimental Setup | Light Intensity | UV Dose (Treatment Time) | Catalyst (Coating Dose) | Coating Surface Area |
|---|---|---|---|---|---|
| [30] | <Lab scale><br>Volume: Not reported<br>Airflow: 200 mL·min$^{-1}$<br>Setup:<br>(1) The targeted gas passed through the surface of a round stainless-steel sample dish with a diameter of 5.5 cm.<br>(2) There was a fan above the lamp and a circulated cooling system. | 0.46 mW·cm$^{-2}$ | Not reported | TiO$_2$<br>(about 1 mg·cm$^{-2}$) | 23.8 cm$^2$ |
| [31] | <Lab scale><br>Volume: Not reported<br>Airflow: <1000 m$^3$·h$^{-1}$<br>Setup:<br>(1) TiO$_2$-coated ceramic filters were used.<br>(2) An exhaust fan was installed at the end of the outlet with a maximum capacity of 7700 m$^3$·h$^{-1}$ to provide a homogenous air stream. | <5.6 mW·cm$^{-2}$ | <1.3 mJ·cm$^{-2}$<br>(<0.32 s) | TiO$_2$<br>(150 cm$^2$·mg$^{-1}$) | 0.65 m$^2$ |
| [32] | <Lab scale><br>Volume: 200 mL<br>Airflow: <300 mL·min$^{-1}$<br>Setup: | 0.06 mW·cm$^{-2}$ | 12 mJ·cm$^{-2}$<br>(< 200 s) | TiO$_2$<br>(10 µg·cm$^{-2}$) | 193.5 cm$^2$ |
| [8] | (1) The standard gases flowing through the UV reactor were irradiated with UV-A.<br>(2) There was a fan above the lamp and a circulated cooling system. | <4.85 mW·cm$^{-2}$ | <0.97 J·cm$^{-2}$<br>(< 200 s) | TiO$_2$<br>(10 µg·cm$^{-2}$) | |

**Table 2.** *Cont.*

| Reference | Experimental Setup | Light Intensity | UV Dose (Treatment Time) | Catalyst (Coating Dose) | Coating Surface Area |
|---|---|---|---|---|---|
| [29] | <Pilot scale, tested at poultry and swine farm><br>Volume: 0.22 m$^3$<br>Airflow: <0.008 m$^3$·s$^{-1}$ | <4.85 mW·cm$^{-2}$ | <0.82 J·cm$^{-2}$<br>(<170 s) | TiO$_2$<br>(10 μg·cm$^{-2}$) | 2.9 m$^2$ |
| [1] | Setup:<br>(1) A fan was installed in a 0.1 m diameter steel axial duct at the end of the reactor. | <0.04 mW·cm$^{-2}$ | <1.88 mJ·cm$^{-2}$<br>(< 47 s) | TiO$_2$<br>(10 μg·cm$^{-2}$) | |
| [25–28] | <Pilot and farm scale, tested at swine farm><br>Volume without walls: 14.4 m$^3$<br>Airflow: <15.1 m$^3$·min$^{-1}$<br>Setup:<br>(1) The mobile laboratory consisted of a series of 12 chambers. Untreated air was brought in through the inlet and treated while flowing in a serpentine pattern from the inlet to the outlet.<br>(2) Fans at entry and near exit of the train of chambers control the airflow rate. | <0.41 mW·cm$^{-2}$ | <5.8 mJ·cm$^{-2}$<br>(<57 s) | TiO$_2$<br>(10 μg·cm$^{-2}$) | 61.2 m$^2$ |
| [24] | <Swine farm, tested at swine farrowing rooms><br>Volume: 455 m$^3$<br>Airflow: < 7666 m$^3$·h$^{-1}$<br>Setup:<br>(1) TiO$_2$ liquid solution was sprinkled on the inside walls of the treated room.<br>(2) In each farrowing room, there were two chimneys. | Not reported | Not reported<br>(<107 s) | TiO$_2$<br>(7 mg·cm$^{-2}$) | 150 m$^2$ |
| [23] | <Swine farm, tested at swine weaning rooms><br>Volume: Not reported<br>Airflow: <10,000 m$^3$·h$^{-1}$<br>Reactor characteristics:<br>(1) TiO$_2$ liquid suspension was sprinkled on the inside walls of the treated unit.<br>(2) Fresh air came from the corridor of the building. The exhaust air was removed by two fans. | <0.45 lux | Not reported | TiO$_2$<br>(7 mg·cm$^{-2}$) | 260 m$^2$ |

Note: < indicates the maximum capacity used in the experiment. It may significantly differ from the dose used to mitigate targeted gases in the actual experiments.

## 4. UV-A Photocatalysis Technology's Effectiveness in Mitigating Targeted Air Pollutants in Livestock and Poultry Barns

In the subsequent tables describing the mitigation results, the reader is encouraged to see the previous experimental conditions and UV doses we calculated from previous studies, which are reported in Table 2 for clarity of comparison.

### 4.1. Mitigation of NH$_3$ and H$_2$S

The reported mitigations of NH$_3$ ranged from 5 to 35% (UV-A dose: 3.9–970 mJ·cm$^{-2}$, Table 3). Statistically significant reductions were observed from lab-scale to farm-scale studies. However, it is difficult to quantitatively evaluate the minimum UV dose and TiO$_2$ coating thickness required to mitigate NH$_3$ in real farm conditions. This is because the information on the irradiance (light intensity) and UV dose was omitted in some works [24,30] that report a relatively high mitigation rate, and the methods used for measuring light intensity were different in other reports. (As noted previously, issues of actual surface geometry could also come into play.) As implied in the Discussion, this does mean that existing data are not of sufficient quality to be able to put a universally meaningful dollar-per-gram price on pollutant removal. In addition, some inconsistencies suggest that not all variables were accounted for. For example, although significant NH$_3$ mitigation was reported on the pilot scale with 5.8 mJ·cm$^{-2}$ of UV dose [26,28], there was no significant mitigation at the farm scale with a similar 5.3 mJ·cm$^{-2}$ of UV dose [27]. It is expected that a higher UV dose is needed on the actual farm due to the harsh environmental conditions (high airborne dust and high relative humidity). Furthermore, higher UV-A

doses and thicker $TiO_2$ coating will likely be necessary to mitigate $NH_3$ efficiently on the farm scale (inside the barn).

**Table 3.** Summary of $NH_3$ and $H_2S$ percent mitigation investigated in previous studies with UV-A photocatalysis. The values reported in the table are statistically significant.

| Reference | Experimental Conditions | UV-A Type (Major Wavelength) | UV Dose (Light Intensity) | Catalyst (Dose) | Gas Mitigation (Mitigation %) |
|---|---|---|---|---|---|
| [30] | Lab scale Temp: 24 °C RH: 50% | Fluorescent (365 nm) | Not reported (0.46 mW·cm$^{-2}$) | $TiO_2$ (approx. 1 mg·cm$^{-2}$) | $NH_3$ (35) |
| [8] | Lab scale (simulated poultry farm) Temp: 25 ± 3 °C RH: 12% | Fluorescent (365 nm) | <88 mJ·cm$^{-2}$ (<0.44 mW·cm$^{-2}$) | $TiO_2$ (10 µg·cm$^{-2}$) | $NH_3$ (9.4) $H_2S$ (N/S) |
| | | LED (365 nm) | <0.97 J·cm$^{-2}$ (<4.85 mW·cm$^{-2}$) | | $NH_3$ (19) $H_2S$ (N/S) |
| [29] | Pilot scale (layer poultry farm) Temp: 28 ± 3 °C RH: 56% | Fluorescent (365 nm) | <75 mJ·cm$^{-2}$ (<0.44 mW·cm$^{-2}$) | $TiO_2$ (10 µg·cm$^{-2}$) | $NH_3$ (5.2) |
| | | LED (365 nm) | <0.82 J·cm$^{-2}$ (<4.85 mW·cm$^{-2}$) | | $NH_3$ (8.7) |
| [28] | Pilot scale (simulated swine farm) Temp: 11 ± 3 °C RH: 34 ± 6% | LED (367 nm) | 3.9 and 5.8 mJ·cm$^{-2}$ (0.41 mW·cm$^{-2}$) | $TiO_2$ (10 µg·cm$^{-2}$) | $NH_3$ (9 and 11) |
| [26] | Pilot scale (simulated swine farm) Temp: 19 ± 2 °C RH: 45 ± 4% | LED (367 nm) | 5.8 mJ·cm$^{-2}$ (0.41 mW·cm$^{-2}$) | $TiO_2$ (10 µg·cm$^{-2}$) | $NH_3$ (6.1) |
| [24] | Swine farm (farrowing rooms) Temp: 24 °C (19–27) RH: 54% | Not reported (315–400 nm) | Not reported | $TiO_2$ (7 mg·cm$^{-2}$) | $NH_3$ (31) |
| [31] | Lab scale (simulated livestock farm) Temp: 20 ± 1 °C RH: 51% | Not reported (368 nm) | 0.6 and 1.3 mJ·cm$^{-2}$ (2.3–5.6 mW·cm$^{-2}$) | $TiO_2$ (1.5 m$^2$·g$^{-1}$) | $H_2S$ (4.2 and 14) |
| [27] | Swine farm (finishing rooms) Temp: 29 ± 2 °C RH: 66 ± 4% | LED (367 nm) | 5.3 mJ·cm$^{-2}$ (0.41 mW·cm$^{-2}$) | $TiO_2$ (10 µg·cm$^{-2}$) | $NH_3$ (N/S) $H_2S$ (26~40) |

Note: Temperature (Temp), relative humidity (RH), not significant (N/S).

The reported mitigation of $H_2S$ ranged from 4 to 40% (UV-A dose: 0.6–5.3 mJ·cm$^{-2}$). As before, it is challenging to make a direct comparison within previous research due to the differences in the $TiO_2$ coating thicknesses, excitation geometries and light intensity measurement. However, an important trend can be noted. In the more controlled lab-scale experiments, there was no statistically significant reduction in $H_2S$ [8], but $H_2S$ mitigation was observed from the more complex mixtures inevitably encountered on the farm scale. It should also be noted that $H_2S$ oxidation via UV-A photocatalysis has been reported multiple times under "clean" laboratory conditions [38,40,53]. This state of affairs clearly needs further investigation.

### 4.2. Mitigation of VOCs and Odor

UV-A photocatalysis can be effective at odorous VOCs mitigation (Table 4). At the lab scale [31,32], some odorous VOCs were effectively removed with a relatively low UV dose compared with the pilot and farm scales. On the pilot scale [1,26,28,29], statistically significant mitigation was reported from a 1.3 mJ·cm$^{-2}$ UV dose. As the UV dose increased,

the VOCs mitigation effect also increased. At the farm scale [27], UV dose $\geq 2.9$ mJ·cm$^{-2}$ partially removed targeted VOCs, and the highest dose (5.3 mJ·cm$^{-2}$) resulted in statistically significant percent mitigation of dimethyl disulfide (62%), isobutyric acid (44%), butanoic acid (32%), *p*-cresol (40%), indole (66%) and skatole (49%). This proves that UV photocatalysis can reduce odorous VOCs even in farm environments.

**Table 4.** Summary of VOCs and odor percent mitigation investigated in previous studies with UV-A photocatalysis. The values reported in the table are statistically significant.

| Reference | Experimental Conditions | UV-A Type (Major Wavelength) | UV Dose (Light Intensity) | Catalyst (Dose) | VOC Mitigation (Mitigation %) |
|---|---|---|---|---|---|
| [31] | Lab scale (simulated livestock farm) Temp: 20 ± 1 °C RH: 51% | Not reported (368 nm) | 0.6 and 1.3 mJ·cm$^{-2}$ (2.3–5.6 mW·cm$^{-2}$) | TiO$_2$ (1.5 m$^2$·g$^{-1}$) | MT (80–87) DMS (92–96) DMDS (83–91) Butan-1-ol (93–95) AA (81–89) PA (97–98) BA (98–99) VA (98–99) |
| [32] | Lab scale (simulated livestock farm) Temp: 40 °C R: 40% | Fluorescent (365 nm) | 12 mJ·cm$^{-2}$ (0.06 mW·cm$^{-2}$) | TiO$_2$ (10 μg·cm$^{-2}$) | DMDS (40) DEDS (81) DMTS (76) BA (87) Guaiacol (100) *p*-Cresol (94) |
| [1] | Pilot scale (swine finishing room) Temp: 22~26 °C RH: 36~80% | Fluorescent (365 nm) | <1.88 mJ·cm$^{-2}$ (<0.04 mW·cm$^{-2}$) | TiO$_2$ (10 μg·cm$^{-2}$) | *p*-Cresol (22) Odor (16) |
| [29] | Pilot scale (layer poultry farm) Temp: 28 ± 3 °C RH: 56% | LED (365 nm) | <0.82 J·cm$^{-2}$ (<4.85 mW·cm$^{-2}$) | TiO$_2$ (10 μg·cm$^{-2}$) | DEDS (47) BA (62) *p*-Cresol (49) Skatole (35) Odor (18) |
| [28] | Pilot scale (simulated swine farm) Temp: 11 ± 3 °C RH: 34 ± 6% | LED (367 nm) | 2.5 and 5.8 mJ·cm$^{-2}$ (0.41 mW·cm$^{-2}$) | TiO$_2$ (10 μg·cm$^{-2}$) | Butan-1-ol (19 and 41) |
| [26] | Pilot scale (simulated swine farm) Temp: 19 ± 2 °C RH: 45 ± 4% | LED (367 nm) | 1.3 and 3.9 mJ·cm$^{-2}$ (0.41 mW·cm$^{-2}$) | TiO$_2$ (10 μg·cm$^{-2}$) | AA (N/S and 49) BA (36 and 53) *p*-Cresol (N/S and 67) Indole (N/S and 32) Odor (N/S and 58) |
| [27] | Swine farm (finishing rooms) Temp: 29 ± 2 °C RH: 66 ± 4% | LED (367 nm) | 2.9 and 5.3 mJ·cm$^{-2}$ (0.41 mW·cm$^{-2}$) | TiO$_2$ (10 μg·cm$^{-2}$) | DMDS (22 and 62) IA (N/S and 44) BA (N/S and 32) *p*-Cresol (32 and 40) Indole (N/S and 66) Skatole (38 and 49) Odor (N/S and 40) |

Note: Temperature (Temp), relative humidity (RH), methanethiol (MT), dimethyl sulfide (DMS), dimethyl disulfide (DMDS), acetic acid (AA), propionic acid (PA), butyric acid (BA), valeric acid (VA), isobutyric acid (IA), not significant (N/S).

The mitigation of odorous VOCs was consistent with the results presented for olfactory odors (16~58%). A statistically significant olfactory odor mitigation was found for higher UV doses, in which the odorous phenolic compounds were mitigated at UV doses of

3.9 mJ·cm$^{-2}$, quite comparable to the 2.9 mJ·cm$^{-2}$ dosages at which several directly targeted VOCs showed reductions on a farm scale [27].

Lee et al. [27] observed a significant change in the perceived overall odor "character" for swine barn emissions after UV-A photocatalysis. The research team described the smell of UV-A photocatalysis treated air as a mix of less-offensive "disinfectant", "minty" or "swimming pool" scents with a weaker smell of swine manure in the background. In addition, the research team described the compound that is believed to have changed the characteristic smell as benzoic acid (or 1-octanol) based on simultaneous chemical and sensory analysis of the GS-MS olfactory test. Although further research on odor character change is still needed, it is interesting that it is the first study to track odor character change after UV-A photocatalysis.

It is important to underline the generation of some targeted compounds for all UV doses in the previous studies [27]. The generated compounds (several in the VFAs group, DMDS and phenol) are odorants that are considered slightly less impactful than *p*-cresol, skatole and indole (representative phenolic compounds), and the generated compounds appear to be partial degradation products from compounds known to be in the original mixtures. Therefore, it is feasible to hypothesize that the generated compounds offset the overall odor mitigation.

### 4.3. Mitigation of GHGs

For the GHGs (Table 5), the previous lab- and pilot-scale study did not find significant mitigation in $CH_4$ under UV-A photocatalysis; however, moderate mitigations (15–27%) were observed with the farm scale and high $TiO_2$ coating thickness [23,24]. It is widely understood from closely related research in photocatalysis in both air and water that hydrocarbons are indeed oxidized (ultimately to $CO_2$) by $TiO_2$ [45,54]; the lack of mitigation in the lab- and pilot-scale reports may reflect insufficient mass transport or other competitive reactions that stop the expected reduction in methane levels. There is little doubt that, in principle, methane should be oxidized.

**Table 5.** Summary of GHGs percent mitigation investigated in previous studies with UV-A photocatalysis. The values reported in the table are statistically significant.

| Reference | Experimental Conditions | UV-A Type (Major Wavelength) | UV Dose (Light Intensity) | Catalyst (Dose) | GHGs Mitigation (Mitigation %) |
|---|---|---|---|---|---|
| [8] | Lab scale (simulated poultry farm) Temp: 25 ± 3 °C RH: 12% | Fluorescent (365 nm) | <88 mJ·cm$^{-2}$ (<0.44 mW·cm$^{-2}$) | $TiO_2$ (10 µg·cm$^{-2}$) | $N_2O$ (3.3) |
| | | LED (365 nm) | <0.97 J·cm$^{-2}$ (<4.85 mW·cm$^{-2}$) | | $CO_2$ (3.8) $N_2O$ (10) |
| [1] | Pilot scale (swine finishing room) Temp: 22~26 °C RH: 36~80% | Fluorescent (365 nm) | <1.88 mJ·cm$^{-2}$ (<0.04 mW·cm$^{-2}$) | $TiO_2$ (10 µg·cm$^{-2}$) | $CO_2$ (−3.1) $N_2O$ (8.7) |
| [29] | Pilot scale (layer poultry farm) Temp: 28 ± 3 °C RH: 56% | Fluorescent (365 nm) | <75 mJ·cm$^{-2}$ (<0.44 mW·cm$^{-2}$) | $TiO_2$ (10 µg·cm$^{-2}$) | $N_2O$ (7.5) |
| | | LED (365 nm) | <0.82 J·cm$^{-2}$ (<4.85 mW·cm$^{-2}$) | | $N_2O$ (13) |
| [26] | Pilot scale (simulated swine farm) Temp: 19 ± 2 °C RH: 45 ± 4% | LED (367 nm) | 2.5 and 3.9 mJ·cm$^{-2}$ (0.41 mW·cm$^{-2}$) | $TiO_2$ (10 µg·cm$^{-2}$) | $N_2O$ (9.0 and 4.3) $CO_2$ (N/S and −25.8) |
| [27] | Swine farm (finishing rooms) Temp: 29 ± 2 °C RH: 66 ± 4% | LED (367 nm) | 2.9 and 5.3 mJ·cm$^{-2}$ (0.41 mW·cm$^{-2}$) | $TiO_2$ (10 µg·cm$^{-2}$) | $N_2O$ (9.4 and 12) $CO_2$ (−33.7 and −27.8) |
| [24] | Swine farm (farrowing rooms) Temp: 24 °C (19–27) RH: 54% | Not reported (315–400 nm) | Not reported | $TiO_2$ (7 mg·cm$^{-2}$) | $CH_4$ (15) $CO_2$ (11) $N_2O$ (4.2) |
| [23] | Swine farm (weaning rooms) Temp: 26 °C (24~30) RH: 56% (52~90) | Not reported (315–400 nm) | Not reported | $TiO_2$ (7 mg·cm$^{-2}$) | $CH_4$ (27) |

Note: Temperature (Temp), relative humidity (RH), not significant (N/S).

A surprising slight decrease in $CO_2$ was reported in a few previous studies [8,24], but most research has reported an increase in $CO_2$ concentration after UV-A photocatalysis [1,25–27,29]. In general, $CO_2$ is the oxidative endpoint for photocatalytic oxidation of virtually all carbon-containing compounds under conditions like those used here. Thus, its mitigation would not derive from its chemical removal [8,29].

$N_2O$ was mitigated by 3–13% under UV-A photocatalysis with 1.9 $mJ\cdot cm^{-2}$ or higher doses [1,8,24–29]. For $N_2O$ concentration, mitigation continuously appeared at a low rate, but the mitigation efficiency did not always increase as the UV dose increased.

In general, $N_2O$ and $O_3$ are known not to absorb significantly in the UV-A range, meaning that they are not subject to direct photolytic degradation at these wavelengths. However, indirect effects through more complex reaction paths can certainly affect their observed concentrations. $N_2O$ might be reduced by the reaction of the hydroxyl radical and activated $O_3$.

### 4.4. Mitigation of Pollutants

Table 6 summarized the additional UV-A photocatalysis mitigation effects other than the gases previously listed.

**Table 6.** Summary of mitigation effects in previous studies with UV-A photocatalysis. The values reported in the table are statistically significant.

| Reference | Experimental Conditions | UV-A Type (Major Wavelength) | UV Dose (Light Intensity) | Catalyst (Dose) | Mitigation (Mitigation %) |
|---|---|---|---|---|---|
| [8] | Lab scale (simulated poultry farm) Temp: 25 ± 3 °C RH: 12% | Fluorescent (365 nm) | <88 $mJ\cdot cm^{-2}$ (<0.44 $mW\cdot cm^{-2}$) | $TiO_2$ (10 $\mu g\cdot cm^{-2}$) | $O_3$ (24) |
| | | LED (365 nm) | <0.97 $J\cdot cm^{-2}$ (<4.85 $mW\cdot cm^{-2}$) | | $O_3$ (48) |
| [26] | Pilot scale (simulated swine farm) Temp: 19 ± 2 °C RH: 45 ± 4% | LED (367 nm) | 1.3 and 5.8 $mJ\cdot cm^{-2}$ (0.41 $mW\cdot cm^{-2}$) | $TiO_2$ (10 $\mu g\cdot cm^{-2}$) | $O_3$ (100 and 100) |
| [29] | Pilot scale (layer poultry farm) Temp: 28 ± 3 °C RH: 56% | Fluorescent (365 nm) | <75 $mJ\cdot cm^{-2}$ (<0.44 $mW\cdot cm^{-2}$) | $TiO_2$ (10 $\mu g\cdot cm^{-2}$) | $O_3$ (100) |
| | | LED (365 nm) | <0.82 $J\cdot cm^{-2}$ (<4.85 $mW\cdot cm^{-2}$) | | $O_3$ (100) |
| [23] | Swine farm (weaning rooms) Temp: 26 °C (24~30) RH: 56% (52~90) | Not reported (315–400 nm) | Not reported | $TiO_2$ (7 $mg\cdot cm^{-2}$) | PM 10 (17) FCR (−12) |
| [27] | Swine farm (finishing rooms) Temp: 29 ± 2 °C RH: 66 ± 4% | LED (367 nm) | 5.3 $mJ\cdot cm^{-2}$ (0.41 $mW\cdot cm^{-2}$) | $TiO_2$ (10 $\mu g\cdot cm^{-2}$) | $O_3$ (100) |
| [25] | Swine farm (finishing rooms) Temp: 29 ± 2 °C RH: 66 ± 4% | LED (367 nm) | 5.3 $mJ\cdot cm^{-2}$ (0.41 $mW\cdot cm^{-2}$) | $TiO_2$ (10 $\mu g\cdot cm^{-2}$) | CFU (49~51) PM (N/S) |

Note: Temperature (Temp), relative humidity (RH), particular matter (PM), feed conversion ratio (FCR), colony-forming unit (CFU), not significant (N/S).

Ozone ($O_3$) is an interesting case because of its well-known atmospheric role in protecting the surface of the earth from UV irradiation. However, $O_3$ does not absorb the light of wavelengths > 290 nm (i.e., UV-A), so its direct photochemistry is not involved in the reported mitigation using UV-A irradiation. Instead, either a reaction with the catalyst or ordinary indirect photocatalytic reactions must be involved [55]. $O_3$ has been reported to increase the reduction in target gas during photocatalysis [56–59] because $O_3$ could be reduced to ozonide radicals ($O_3^-$). In this instance, $O_3$ would be an electron sink (in parallel with ambient $O_2$), but the resulting ozonide is sufficiently reactive that the ozone is destroyed rather than reformed by oxidation.

At both farm and pilot scales, ambient $O_3$ was removed completely, whereas at the lab scale, mitigation was significant but not complete (24–48%). It is reasonable to speculate that a wider variety of compounds are produced in the larger-scale reactions than in the "controlled" lab reactions and that some of these are reactive with ozone or ozonide.

The effects of UV-A photocatalysis on PM have been investigated by two groups. An Italian research team [23] reported that UV-A photocatalysis with $TiO_2$ mitigated airborne PM 10, which is a PM with diameters that are usually 10 microns or less (17%) inside a ~390 head nursery barn, while also improving feed conversion efficiency (12%). The mechanism of PM mitigation was not reported and was unclear; thus, the Iowa (USA) research team independently investigated the PM mitigation effect. No statistically significant PM mitigation was reported in experiments that used different minimum efficiency reporting values (MERV) rating filters to create three different airborne PM concentration levels [25].

Although pathogens are obviously much larger than individual gas molecules, it is well known in other applications of photocatalytic methods that they can be inactivated without the need for complete chemical degradation. The research of Rodriguez-Silva et al. [60] is an example of microorganism inactivation in liquid water. The previous precedent showed that microbe deactivation by photocatalysis is sensitive to catalyst loading and UV dose, like chemical degradation. Therefore, it is considered that there is a potential to inactivate airborne microorganisms if appropriate UV-A dose and $TiO_2$ coating are satisfied. Indeed, UV-A photocatalysis treatment was reported to mitigate airborne microbial colony-forming units (CFUs, a measure of the airborne microbial load) by 15–95% [25]. Normalization of the measured airborne pathogen concentrations by smaller PM size concentrations led to the significant mitigation (49–51%, *p*-value < 0.03) effect of UV-A photocatalysis on pathogen inactivation [25].

**5. Discussion**

*5.1. Expected Advantage of UV-A Photocatalysis in the Livestock and Poultry Barn*

The mitigation of air pollutants by UV-A photocatalysis in livestock and poultry farming is reported in Table 7 based on the published research papers to date. Conflicting mitigation results of $H_2S$, $CH_4$ and PM in previous papers were excluded from Table 7. Considering the reported mitigation effect in the farm environment, UV-A photocatalysis could be expected to mitigate airborne microbials and improve the air quality.

**Table 7.** Summary of UV-A photocatalysis mitigation under farm conditions in previous studies.

| Reference | Mitigated Gas (Maximum Mitigation [1] %) | Minimum UV Dose [2] | Catalyst (Dose) |
|---|---|---|---|
| [24] | $NH_3$ (~35%) | Not reported | $TiO_2$ (7 mg·cm$^{-2}$) |
| [27] | VOCs (~66%) | 2.9 mJ·cm$^{-2}$ | $TiO_2$ (10 μg·cm$^{-2}$) |
| [27] | Odor (~58%) | 4.0 mJ·cm$^{-2}$ | $TiO_2$ (10 μg·cm$^{-2}$) |
| [24,26–29] | $N_2O$ (~13%) | 1.9 mJ·cm$^{-2}$ | $TiO_2$ (10 μg·cm$^{-2}$) |
| [26–29] | $O_3$ (~100%) | 1.3 mJ·cm$^{-2}$ | $TiO_2$ (10 μg·cm$^{-2}$) |
| [25] | Airborne microbials (~51%) | 5.3 mJ·cm$^{-2}$ | $TiO_2$ (10 μg·cm$^{-2}$) |

[1] The highest mitigation rate reported under farm scale in previous studies. [2] The minimum UV dose with a statistically significant reduction in the targeted gas under farm scale.

These positive advantages of UV-A photocatalysis are considered beneficial to animal health and welfare. To help develop animal welfare standards, the World Organization for Animal Health (OIE) released 10 "General Principles for the Welfare of Animals in

Livestock Production Systems" [61]. Of these 10 principles, UV technology could help enhance the following principles:

"(2) how the environment influences injuries and the transmission of diseases and parasites";

"(5) the effects of air quality, temperature and humidity on animal health and comfort";

"(6) ensuring access to feed and water suited to the animals' needs and adaptations".

As shown in the previous sections, UV-A photocatalysis can have a role in each of these principles. Ni et al. [62,63] summarized some observational studies that confirmed air toxicity could result in animal deaths or injuries. The reported primary pollutants are $NH_3$, PM, $H_2S$, bacteria and endotoxins. UV-A photocatalysis can contribute to mitigating part of the pollutants to improve animal welfare. The role of UV-A photocatalysis in improving the feed conversion rate is more speculative, but obviously, the general health of the livestock from improved air quality is a contributing factor here.

Although UV-A photocatalysis under "farm conditions" shows promising mitigation effects, it should be clearly recognized that the UV dose required for significant mitigation of odorous gas cannot be determined through studies conducted to date. As noted previously, most of the UV-A photocatalytic reactions occur at or very near the coated surface. This fact implies that the physical geometry of the experimental setup also has measurable effects on the rate/efficiency of degradation. Thus, the experimental chambers used in these previous studies may not be optimized, even granting that they were well thought out. In addition, all available surfaces should be coated with a catalyst to maximize the efficiency of the apparatus, but this complicates quantitative analysis due to the varying light intensity. Furthermore, the actual coating thickness of $TiO_2$ is a very small part of the overall cost and may not have been optimized in the reported works. Increasing the total UV dose can be accomplished by manipulating the airflow or increasing the total number of bulbs; of course, this variable represents an ongoing cost. The changes in airflow can be difficult to implement in practice because mechanical barn ventilation is driven by indoor air temperature and relative humidity considerations.

*5.2. Improvements Needed to UV-A Photocatalysis in Livestock and Poultry Barns*

The currently needed improvement for quantitative analysis is that a unified index reflecting the characteristics of UV-A photocatalysis is needed to evaluate and compare the efficiency of UV-A photocatalysis between different studies. UV photocatalysis is a reaction that takes place on a surface coated with a photocatalyst, unlike direct UV photolysis. In direct photolysis, all targeted gases are treated under the same or higher UV dose than the UV dose measured from the opposite side of the installed lamp, in that the light is partially absorbed by the contents of the apparatus in the irradiation path. Very reasonable estimates of the absorbed energy could be obtained by measuring the intensity adjacent to the light sources and at the end of the path length. However, photocatalytic reactions are activated by the light intensity hitting the surface in combination with the gas having contact and near contact with the surface. In a "real" (i.e., not ideal) apparatus, the UV dose on the coated surface varies greatly depending on the position of the surface relative to the lamps. Truly quantitative comparisons would require an integration of UV doses across all surfaces.

The UV dose in the previous papers was either (1) measured on the surface irradiated directly opposite the UV-A lamp or (2) averaged over all interior walls coated with catalyst by the UV-A lamp. As an example, Lee et al. [28] used the all-surface (three-dimensional) method to measure the light intensity, using averaged intensities and surface areas. Thus, the measured UV dose was 5.8 mJ·cm$^{-2}$, but if the UV dose was calculated using the opposite-wall-only method, it would increase to ~0.12 J·cm$^{-2}$ because that was the measured value on the most directly irradiated surface.

The index we propose is to measure the UV dose on all coated interior walls rather than only on the walls directly opposite the lamps and then to estimate the amount of energy (J) over the total coated surface area. This can be adjusted for the volume of the gas and/or pollutant concentration inside the apparatus if the mitigation is measured

in a relative manner, such as percent, instead of an absolute manner, such as g or moles. Given the currently available data, the best possible comparison is given in Table 8. For studies performed under different surface areas and conditions, the total UV energy (mJ) was estimated by multiplying the UV dose (mJ·cm$^{-2}$) by the total coated surface area (cm$^2$). Table 8 shows the UV energy needed to mitigate the targeted gas by reflecting the surface area used for the experiment with a UV dose from Tables 1–7. In previous lab-scale experiments, an average of 66 J (range: 2–188 J) was irradiated to the TiO$_2$ coated surface for significant targeted gas mitigation, and an average of 9128 J was required in farm-scale experiments (range: 55 J–23,780 J).

**Table 8.** The energy of UV-A photocatalysis needed for mitigating targeted gases in previous studies.

| Reference | Mitigated Gases (Maximum Mitigation %) | Light Intensity | UV Dose (Treatment Time) | Catalyst (Coating Dose) | Coating Surface Area | UV Energy |
|---|---|---|---|---|---|---|
| [30] | NH$_3$ (35) | 0.46 mW·cm$^{-2}$ | Not reported | TiO$_2$ (~1 mg·cm$^{-2}$) | 23.8 cm$^2$ | Not available |
| [31] | H$_2$S (14), MT (87), DMS (96), DMDS (91), Butan-1-ol (95), AA (89), PA (98), BA (99), VA (99) | 5.6 mW·cm$^{-2}$ | 1.3 mJ·cm$^{-2}$ (<0.32 s) | TiO$_2$ (150 cm$^2$·mg$^{-1}$) | 0.65 m$^2$ | 8.45 J |
| [32] | DMDS (40), DEDS (81), DMTS (76), BA (87), Guaiacol (100), *p*-Cresol (94) | 0.06 mW·cm$^{-2}$ | 12 mJ·cm$^{-2}$ (<200 s) | TiO$_2$ (10 µg·cm$^{-2}$) | 193.5 cm$^2$ | 2.33 J |
| [8] | NH$_3$ (19), N$_2$O (10), O$_3$ (48) | <4.85 mW·cm$^{-2}$ | <0.97 J·cm$^{-2}$ (<200 s) | TiO$_2$ (10 µg·cm$^{-2}$) | | 188 J |
| [29] | NH$_3$ (8.7), DEDS (47), BA (62), *p*-Cresol (49), Skatole (35), Odor (18), N$_2$O (13), O$_3$ (100) | <4.85 mW·cm$^{-2}$ | <0.82 J·cm$^{-2}$ (<170 s) | TiO$_2$ (10 µg·cm$^{-2}$) | 2.9 m$^2$ | 23,780 J |
| [1] | CO$_2$ (−3.1), N$_2$O (8.7), *p*-Cresol (22), Odor (16) | <0.04 mW·cm$^{-2}$ | <1.88 mJ·cm$^{-2}$ (<47 s) | TiO2 (10 µg·cm$^{-2}$) | | 54.5 J |
| [25–28] * | NH$_3$ (11), H$_2$S (40), DMDS (62), IA (44), BA (32), Indole (66), Skatole (49), Butan-1-ol (41), Odor (40), CO$_2$ (−27.8), N$_2$O (12), O$_3$ (100), CFU (51), | <0.41 mW·cm$^{-2}$ | <5.8 mJ·cm$^{-2}$ (<57 s) | TiO$_2$ (10 µg·cm$^{-2}$) | 61.2 m$^2$ | 3550 J |
| [24] | NH$_3$ (31), CH$_4$ (15), CO$_2$ (11), N$_2$O (4.2) | Not reported | Not reported (<107 s) | TiO$_2$ (7 mg·cm$^{-2}$) | 150 m$^2$ | Not available |
| [23] | CH$_4$ (27), PM (17), FCR (−12) | <0.45 lux | Not reported | TiO$_2$ (7 mg·cm$^{-2}$) | 260 m$^2$ | Not available |

\* The highest mitigation rate reported in previous studies. Methanethiol (MT), dimethyl sulfide (DMS), dimethyl disulfide (DMDS), acetic acid (AA), propionic acid (PA), butyric acid (BA), valeric acid (VA), isobutyric acid (IA), particular matter (PM), feed conversion ratio (FCR), colony-forming unit (CFU).

For the sustainable mitigation effect of UV-A photocatalysis, it is recommended that proper PM management should be considered simultaneously with UV-A photocatalysis on the farm. UV photocatalysis causes a change in the targeted material through the chemical activity of the catalyst coated on the surface with irradiated UV light. Thus, PM accumulation on TiO$_2$-coated surfaces that block the transmission of the light (and gases) to the catalyst surfaces could affect treatment effectiveness. Lee et al. [28] reported that PM accumulates on the TiO$_2$ surface if not cleaned. In addition, Lee et al. [25–27] reported that the mitigation effect of UV-A photocatalysis decreased as the amount of airborne PM increased. Similarly, PM accumulation on the lamps would decrease performance. In general, a combination of PM management and periodic cleaning of lamps and surfaces is expected to be required to maintain the highest efficiency. However, Zhu et al. [32] reported that PM accumulation in the swine house did not significantly affect odorous VOCs mitigation. Therefore, further investigation of accumulated PM effects on the photocatalyst surface is necessary.

Further research on the physical stability of TiO$_2$ coatings in farm conditions will be needed. While TiO$_2$ coatings are robust to humid air and water, Lee et al. [28] reported that the TiO$_2$ coated on a glass surface was completely removed with propan-2-ol (isopropyl alcohol; see Figure A9.e in Ref [28]). Another work suggests that TiO$_2$ can operate under high-humidity conditions for an extended period of time [29]. However, it is considered

that additional experiments are required to test the practical application of TiO$_2$ coating inside farms where power washing with water (and sometimes with disinfectants) is performed periodically or in environments where condensation is formed on the wall and ceiling due to temperature differences inside and outside. The material cost of TiO$_2$ is sufficiently low that the labor and other practical issues surrounding re-coating would be a greater factor than the catalyst material itself.

It needs to be recognized that the reported data, while promising, are not yet prescriptive for widespread commercialization, at least in part due to inconsistencies in the measurement techniques, which differed among the reports. Because of the complexity of farm air atmospheres (fast-moving air, dust, etc.), truly accurate measurement of light dosage cannot simply come from the wattage of lamps or, in the absence of "real" exhaust, through the system with light intensities measured at the surfaces where the catalyst absorbs the light. Clearly, many geometries of actual farm systems would not have light as ideally distributed as some of the systems described in previous research. Designs that maximize the efficiency of light usage and gas-surface contact will be more efficient from at least that standpoint. Computational fluid dynamics (CFD) modeling may play a role in improving the design of UV-A irradiation chambers by optimizing the coating areas, airflow rate, interior geometry and UV lamp positions.

*5.3. Cost Effectiveness of UV-A Photocatalysis Application in the Livestock and Poultry Housing*

The currently available data are insufficient to evaluate the economic analysis of UV-A photocatalysis. Only a few studies assessed the economic feasibility by comparing the cost of electricity consumed for quantitatively mitigating targeted gas (per gram) in a farm environment. Readers should be aware that the economic feasibility analysis in the previous studies is subject to the efficiencies and variables noted above.

Current data suggest that UV-A photocatalysis is not a cost-effective method to mitigate NH$_3$ or H$_2$S on the farm scale. According to Lee et al. [26–29], mitigating 1 g of NH$_3$ costs about USD 0.70~3.50.; removing 1 g of H$_2$S costs USD 1.09. NH$_3$ and H$_2$S are emitted in tens of grams to kilogram quantities from the swine farms each day [64]. The currently estimated costs of N$_2$O (emitted in mg–g per day) and O$_3$ mitigation are relatively low at < USD 10 per day [27,29]. However, it cannot be said that these values are economically reasonable. In addition, the cost effectiveness of UV-A photocatalysis was not thoroughly evaluated, as there is no research where the estimated economic feasibility of mitigating odorous VOCs and odor, the biggest advantage of UV-A photocatalysis, has been analyzed.

In conclusion, the current estimates suggest that UV-A photocatalysis is not yet an economically viable technology on the farm, as at the time of writing of this article, though its potential cannot yet be ruled out. Approaching it from a slightly different angle, for example, light irradiation is required purposefully for the reproductive cycle in a poultry barn. Therefore, it could be a net benefit if only the light bulb could be replaced with a UV-A lamp to produce a photocatalytic advantage while maintaining the original electricity cost. The proof of economic viability clearly awaits either more sophisticated cost estimates that show the current values to be wrong or more efficient photocatalytic devices.

*5.4. Possible Future Trends in UV-A Photocatalysis Applications for Livestock Facilities*

UV-A photocatalytic technology has great potential, considering its demonstrated ability to oxidize odorous gases, inactivate airborne microbes and potentially improve the feed conversion ratio as a secondary effect. However, the currently available data fail to establish that the method is economically viable for farm applications. On a positive note, technological advances and continuous research may improve the mitigation efficiency per dollar. For example, the filtered far UV-C excimer lamp (207–222 nm), which is claimed not to cause skin burns and eye injuries, is being developed commercially [65]. Photocatalysis can also be made more cost efficient by improving the photonic efficiency of the catalyst itself. Although not addressed directly in this review, the fraction of photons absorbed by TiO$_2$ that is converted into an actively used h$^+$/e$^-$ pair is relatively small, and finding more

efficient catalysts, particularly those that can take advantage of photons in the visible light regions, is an active area of research.

Another future potential value of UV photocatalysis may come from the antibacterial activity of photolysis and/or photocatalysis. Although farms are systematically designed to meet the needs of livestock at each stage of growth, several factors (spilled feed, excreted manure, odorous gases and flies) can create a disease-prone environment without proper management. Disease outbreaks on farms cause substantial economic loss. The bactericidal effect from UV photocatalysis may attract attention as an application to prevent this loss in the future. For example, UV-A photocatalysis has shown bactericidal effects for bioaerosols [66,67], surfaces [68] and liquids [69,70].

In terms of mitigation and inactivation of indoor pollutants, photocatalytic pollutant mitigation could be more practical for interior applications (Figure 2) than for open settings, such as a cattle feedlot. Future developments of UV-photocatalysis are clearly required to establish it as an economical tool for farm use. Nonetheless, as the studies discussed in this review show, there is a clear proof of concept for the improvement of air quality, workers' health, animal welfare and reduced environmental impact of the farm.

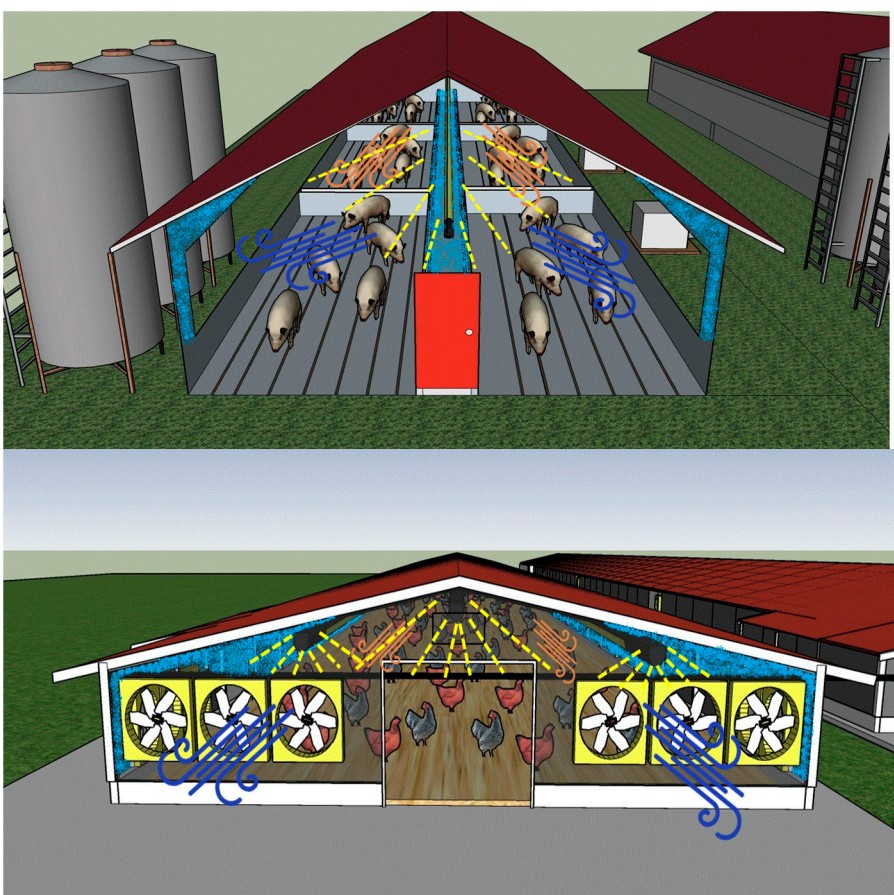

**Figure 2.** A conceptual illustration of swine and poultry barns with UV photocatalysis in the future. Photocatalyst (bright blue), UV light (yellow rays), treated air (brown to dark blue). Note that UV-A light needs to be shielded, or the lighting duration needs to be controlled, as UV-A overexposure has side effects.

## 6. Materials and Methods

Google Scholar was used to search for the relevant published literature. Initially, the topic "gas mitigation" yielded nearly 2,280,000 documents, and then, the screened topic yielded 261 documents by adding the sub-topic "UV-A" and "Farm". The 261 documents enabled us to narrow down around 80 papers that were clearly within the scope of this

review topic, which was the mitigation of air pollutants by UV-A photocatalysis in livestock and poultry farming.

The 80 papers related to this review topic were organized and summarized based on three categories:

(1)    the mitigation method with UV light (photolysis and photocatalysis);
(2)    the mitigation rate of odorous gases;
(3)    coating thickness of catalyst and UV dose.

The summarized data were reorganized into tables and contents in this review paper. The overall review method is shown in Figure 3.

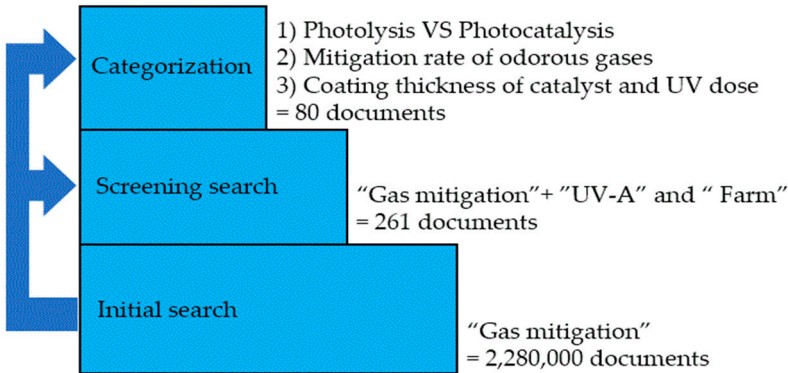

**Figure 3.** Methodology of literature search used.

## 7. Conclusions

UV-A photocatalysis has been evaluated for mitigation of odorous gases, airborne microbials and potentially increased feed conversion rates. Relatively fewer works have reported on applications beyond the proof-of-the-concept scale. Still, little is known about the current state of the art of UV-A photocatalysis in the context of animal agriculture. This review paper aimed to present the UV-A photocatalyst mechanism, summarize UV-A photocatalysis's effectiveness in mitigating targeted air pollutants in livestock and poultry barns, and evaluate the potential of UV-A photocatalysis. We traced the progression of UV-A photocatalysis applications in animal farming since the mid-2000 and developments from laboratory- to farm-scale trials. In addition, this review paper discussed the practical limitations and outlined the research needs for increasing the technology readiness and practical UV application in animal farming.

**Author Contributions:** Conceptualization, M.L. and J.A.K.; methodology, M.L. and J.A.K.; resources, J.A.K.; writing—original draft preparation, M.L., J.A.K., P.L. and W.S.J.; writing—review and editing, M.L., J.A.K., P.L. and W.S.J.; visualization, M.L.; supervision, J.A.K.; project administration, J.A.K.; funding acquisition, J.A.K. All authors have read and agreed to the published version of the manuscript.

**Funding:** J.A.K.'s participation (while still primarily affiliated with Iowa State University) was partially supported by the Iowa Agriculture and Home Economics Experiment Station, Ames, Iowa. Project no. IOW05556 (Future challenges in animal production systems: Seeking solutions through focused sponsored by Hatch Act & State of Iowa funds).

**Data Availability Statement:** The original contributions presented in the study are included in the article; further inquiries can be directed to the corresponding author.

**Conflicts of Interest:** The authors declare no conflict of interest. The funders did not play any role in the study design, data collection, analysis, interpretation, and decision to write a manuscript or present the results.

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
