# Peer review of "Mitigation of Air Pollutants by UV-A Photocatalysis in Livestock and Poultry Farming: A Mini-Review"

_catalysts, doi:10.3390/catal12070782_

Round 1
Reviewer 1 Report
The manuscript entitled with “Mitigation of air pollutants by UV-A photocatalysis in live- 2 stock and poultry farming: a mini-review” summarized UV-A photocatalysts for air pollutants purification. The whole construction is well summarized and this review can be accepted after following minor revisions:
(1) It’s better to give some scheme’s figures for different contents in section 2 and 3 other than tables.
(2) This referee notes that most of references are old, it’s better to add some new references within last 3 years.
(3) The look into the future and shortcomings are need in the last section 5.
Reviewer 2 Report
1. The literature study must be enriched. indeed, the following all of references about photocatalysts are recommended to be cited: Green synthesis of DyBa2Fe3O7. 988/DyFeO3 nanocomposites using almond extract with dual eco-friendly applications: Photocatalytic and antibacterial activities// Synthesis, characterization and application of Co/Co3O4 nanocomposites as an effective photocatalyst for discoloration of organic dye contaminants in wastewater and antibacterial properties///Green sonochemical synthesis of BaDy2NiO5/Dy2O3 and BaDy2NiO5/NiO nanocomposites in the presence of core almond as a capping agent and their application as photocatalysts for the removal of organic dyes in water.
2. The relevance/novelty of the work needs to be highlighted.
3. Check the grammar throughout the article and correct it. Proofread the article as many language errors were identified.
Reviewer 3 Report
The manuscript presents a mini-review on the application of ultraviolet-based treatment technologies for the treatment of poultry farming air pollutants. The manuscript can be considered for publication after the following modifications.
1. What is UV-A? Please provide the full abbreviation.
2. The paper is a review but has the format of an original paper. The results section is not meaningful in review papers!
3. The paper fails to discuss the possible future trends in the application of such technologies. I suggest including a section in the manuscript to discuss this.
4. Cost-effectiveness of such technologies for the commercialization of such technologies has not been discussed in the manuscript.
Round 2
Reviewer 3 Report
No more specific comments!